# ARDformer: Agroforestry Road Detection for Autonomous Driving Using Hierarchical Transformer

**DOI:** 10.3390/s22134696

**Published:** 2022-06-22

**Authors:** Eksan Firkat, Jinlai Zhang, Danfeng Wu, Minyuan Yang, Jihong Zhu, Askar Hamdulla

**Affiliations:** 1College of Information Science and Engineering, Xinjiang University, Urumqi 830049, China; eksan@stu.xju.edu.cn (E.F.); jhzhu@tsinghua.edu.cn (J.Z.); 2College of Mechanical Engineering, Guangxi University, Nanning 530004, China; 1711302013@st.gxu.edu.cn; 3College of Robotics, Beijing Union University, Beijing 100100, China; jqrdanfeng@buu.edu.cn; 4School of Vehicle and Mobility, Tsinghua University, Beijing 100190, China; yang-my19@mails.tsinghua.edu.cn; 5Department of Precision Instrument, Tsinghua University, Beijing 100190, China

**Keywords:** road detection, agroforestry, semantic segmentation, transformer, scene understanding

## Abstract

Road detection is a crucial part of the autonomous driving system, and semantic segmentation is used as the default method for this kind of task. However, the descriptive categories of agroforestry are not directly definable and constrain the semantic segmentation-based method for road detection. This paper proposes a novel road detection approach to overcome the problem mentioned above. Specifically, a novel two-stage method for road detection in an agroforestry environment, namely ARDformer. First, a transformer-based hierarchical feature aggregation network is used for semantic segmentation. After the segmentation network generates the scene mask, the edge extraction algorithm extracts the trail’s edge. It then calculates the periphery of the trail to surround the area where the trail and grass are located. The proposed method is tested on the public agroforestry dataset, and experimental results show that the intersection over union is approximately 0.82, which significantly outperforms the baseline. Moreover, ARDformer is also effective in a real agroforestry environment.

## 1. Introduction

Using agroforestry robots can lower production costs and reduce manual labor demand, thereby significantly improving production efficiency. However, as a fundamental part of the robotic perception system, road detection in agroforestry environments is studied relatively less than road detection in urban streets. The urban roads have clear lane lines and traffic signs. However, the agroforestry environment is mainly unstructured, thus making road detection in agroforestry environments challenging. Road detection in agroforestry environments is more challenging than urban roads.

The existing approaches for road detection fall into two categories: the rule-based method and segmentation-based method [1]. Rule-based methods assume some global road priors/constraints, such as vanishing points and geometric shape [2,3]. However, this approach does not perform well without distinct boundaries, and the diversity of road surfaces makes rule-based approaches error prone. The segmentation-based method converts road detection to extract the continuous region based on a specific pixel-level road model [4]. Semantic segmentation has recently become the most popular method for this task with the help of deep learning [5]. Most semantic segmentation methods segment the scene based on specific descriptive categories. However, such descriptive categories are not directly definable for road detection in agroforestry [6].

As illustrated in Figure 1b, due to the limitation of descriptive categories of the scene, when a trail surrounds grass, the segmentation model neglects the grass and segments the trail as a road. Additionally, most of the segmentation models are FCN-based methods [7]. The locality property of CNN limits the long-range dependency between different areas and degrades the performance of semantic segmentation [8]. A traditional rule-based method, such as vanishing point detection shown in Figure 1c, constrains the road region in agroforestry as a triangle and violates the distribution of roads in agroforestry. The rule-based method only works well when road boundaries are clear. Moreover, as shown in the bottom row of Figure 1c, rule-based methods are prone to errors if the distribution of roads is diverse.

This paper addresses the problem mentioned above by a novel multi-stage road detection approach. Specifically, we propose a novel agroforestry road detection algorithm based on the hierarchical transformer, ARDformer. ARDformer is a two-stage approach: a densely connect hierarchical transformer is used in the first stage to segment the agroforestry environment, then the scene mask is extracted for road estimation in the next stage. In order to test the proposed method, we conducted several experiments on agroforestry dataset, the FreiburgForest [9], and results showed that our ARDformer outperforms other methods significantly. In addition, to validate the robustness of ARDformer, we also experimented with our approach on another agroforestry dataset and a real agroforestry environment. The experimental results showed that our ARDformer is also effective in those scenarios. The road detection results are shown in Figure 1d.

This work provides the following contributions: (1) We propose a novel road detection algorithm that solves the descriptive categories that are not directly definable for road detection in agroforestry. (2) We introduce a novel transformer-based semantic segmentation method for road detection and outperform the baseline. (3) The proposed ARDformer can be applied to various agroforestry environments.

The remainder of this paper is organized as follows: First, the related works are introduced in Section 2. Section 3 presents the proposed methodology in detail. Then Section 4 shows experimental results. Finally, we draw some conclusions in Section 5.

## 2. Related Work

### 2.1. Road Detection

The road detection problem has been studied intensively in recent years. Although 3D sensors, such as laser or LiDAR, provide good geometry information, their sparsity brings difficulty to road detection [10]. Therefore, most of the methods are concerned with vision-based methods and can be broadly categorized as rule based and segmentation based.

The rule-based method assumes that certain global priors/constraints, such as vanishing points and geometric shapes, can be found in the image. The vanishing point has been studied extensively for road detection. There are two standard processes for vanishing point road detection. First, the vanishing point is associated with the central part of the road. Then road areas are segmented based on the detected vanishing point [2]. The estimation of the vanishing point plays a critical role in this method. Rasmussen [11] introduced the representative work for vanishing point estimation for road detection. He utilized 72-oriented Gabor filter templates to acquire the vanishing point. The rest of the studies mostly improve accuracy and reduce runtime. Kong et al. [2] and Moghadam et al. [12] optimized the filter bank to improve the accuracy of the vanishing point and speed up the estimation of the vanishing point. The vanishing point method depends on edge cues, and when the scene contains well-defined edges, the vanishing point approach can work well. Otherwise, the robustness considerably degrades. Road region detection methods based on certain geometric shapes (such as triangle, trapezoid or linear boundary constraints) have also been widely studied. Kong et al. [2] constrained the road surface as triangular, then one vanishing point and two road boundaries were detected to form a triangular road region. However, the triangle constraint is unsuitable for curved roads and intersections in off-road scenes. Rule-based methods attempt to use the global constraint to detect the road in an off-road scene. However, this approach does not work as expected when there are no clear borders, and the variety of road surfaces makes rule-based methods prone to errors.

The segmentation-based method formulates the problem as pixel-level classification tasks. Lu et al. [13] considered the region at the bottom of images as road data. Mei et al. [1] collected the vehicle trajectories as drivable areas and segmented the similar region as roads. Wang et al. [14] and Lu et al. [13] detected the road region based on fixed road models and hybrid features. In recent years, large-scale datasets, such as cityscapes and SemanticKITTI [15,16], made deep learning approaches successfully applied to various robotic vision tasks, including object recognition, detection, and semantic segmentation [5]. The semantic segmentation method has been introduced to the urban scene road detection task [17]. However, there is no large-scale dataset in agroforestry environment, and most semantic understanding of the scene is based on specific descriptive categories, which causes the definition of road regions in agroforestry to remain challenging.

The rule-based detection approach ignores the “region-based” nature of the road and only focuses on detecting edge curves of the road, whereas the semantic-based approach lacks a geometric model of the road, which can be described by a small number of parameters assuming that the road is planar and viewed by a perspective projection. This paper combines the advantage of two methods and proposes a novel road detection approach for agroforestry. We combine semantic scene segmentation with road estimation to detect road regions in an agroforestry environment.

### 2.2. Transformer

For natural language processing (NLP), Transformer [18] achieves state-of-the-art performance. Tremendous breakthroughs in the natural language domain inspire researchers to explore the potential and feasibility of Transformer in the computer vision field [8]. Recently, Transformer has been gaining increasing interest in many computer vision tasks [19].

Unlike the convolutional neural networks (CNN) [20,21] models’ local image pattern, Transformer employs self-attention [22] on a tokenized image patch to model the context information of the image. In recent studies, pure Transformer architectures were introduced to many visual recognition and image classification tasks and achieved promising results. Carion et al. [23] proposed an end-to-end transformer for object detection, which passes CNN features to Transformer, and the object class and location are generated by Transformer. Wang et al. [24] proposed a dual-branch transformer for panoptic segmentation, where the memory branch is responsible for class prediction, and the pixel branch is responsible for segmentation. Ding et al. [25] took advantage of both CNN and Transformer for the semantic segmentation of remote sensing images. CNN is good at spatial information, and Transformer enables the better modeling of long-range dependencies. The proposed dual branch context transformer model has broader context in images. Chang et al. [26] proposed U-net structure transformer called TRANSCLAW U-NET for medical image segmentation. The encoding part is composed of a convolution operation and transformer operation to obtain the feature representation of image.

The pioneering work of the swin-transformer presents a hierarchical feature representation scheme that demonstrates impressive performances with linear computational complexity. This paper utilizes the swin-transformer as the backbone for agroforestry scene segmentation.

## 3. Methodology

The overall framework of our ARDformer is shown in Figure 2. In the semantic segmentation stage, the proposed densely connected hierarchical Transformer segments the input image to scene mask. Once a scene mask is obtained, the road region is detected by the road estimation stage. First, the edge detection algorithm detects the edge of the trail, then the road region is detected by the periphery estimation. The road area refers to the area enclosing the perimeter of a trail that contains a trail and grass.

### 3.1. Densely Connect Hierarchical Transformer

The overall structure of the densely connected hierarchical transformer is shown in Figure 3, which includes swin-transformer [8] as the feature extraction backbone and hierarchical aggregation module (HAM) for decoding the image.

Swin-transformer constructs a hierarchical feature map with linear computational complexity to image size. First, the input image is split as non-overlapping patches, and each patch’s feature is set as the concatenation of the RGB value. Patches are then fed into a multi-stage transformer for feature transformation. In the first transformer stage, patches are fed into the linear embedding layer and projected to an arbitrary dimension C. Once the projection operation is complete, patches are fed into the transformer block for feature transformation. The transformer block consists of a shifted window-based multi-head attention module, followed by a 2-layer MLP with GELU [27] nonlinearity. In the remaining transformer stage, the number of patches is gradually reduced by the patch merging operation to produce a hierarchical representation of the input image. Each group of neighboring 2 × 2 patches is concatenated in the remaining stage as a new patch, followed by a transformer block to feature transformation. Proceed by four stages, four hierarchical representations of different sizes are generated.

To fully exploit the swin-transformer’s [8] hierarchical representation, the hierarchical aggregation module (HAM) is proposed for feature representation. Specifically, multi-level features are further integrated using hierarchical connection and skip aggregation to improve the multi-scale representation. As shown in Figure 3, the hierarchical connection connects the four hierarchical transformer features with cross-scale connections, generating four aggregated features. The aggregate feature is used for final segmentation.

The HAM aims to connect the low-level and high-level transformer feature, which can be formalized as
(1)$$H\left(M\right)=Relu(f_{2}^{2\times 2}\left(M\right)+f_{2}^{2\times 2}\left(f_{2}^{2\times 2}\left(M\right)\right))$$
where *M* is the input tensor, Relu [28] refers the rectified linear activation function, $f_{1}$ represents a convolution operation with the filter size of 7×7 and stride of 2, and $f_{2}$ is stander 1×1 convolution operation. Moreover, to capture the hierarchical feature effectively, the dilated convolution operation is embedded into skip aggregation as
(2)$$S\left(M\right)=D_{2}\left(Relu\left(D_{2}\left(M\right)\right)\right)$$
where $D_{1}$ is a dilate convolution operation with dilated rate of 12. Similarly $D_{2}$ has a dilate rate of 6. In addition, the four hierarchical feature can eventually be computed by the following operation:(3)$$AF^{4}=ST^{4}+H\left(AF^{2}\right)$$
(4)$$AF^{3}=ST^{3}+H\left(H\left(AF^{1}\right)\right)$$
(5)$$AF^{2}=ST^{2}+H\left(H\left(AF^{4}\right)\right)$$
(6)$$AF^{1}=ST^{1}+U\left(AF^{2}\right)+S\left(AF^{2}\right)$$
where *U* is a bilinear interpolation upsample operation with a scale factor of 2. Capitalizing on the benefits provided by cross-scale connections, the final segmentation feature $AF^{1}$ is abundant in multi-scale and contextual information.

### 3.2. Road Estimation

Figure 4 illustrates the process of estimating road in agroforestry using the scene mask of the trail. Edge points of the trail are selected to estimate the periphery of the trail, and a simple and intuitive method is used to compute periphery of the trail. Firstly, the trail in the scene mask is extracted based on color representation of the trail, and edge points of the trail are detected using canny edge detection algorithm. Then edge points of the trail are selected to detect the road in agroforestry. The road detection algorithm is used for periphery estimators to enclose the trail region where those areas can be determined as roads. Details of the road detection are summarized in Algorithm 1.
**Algorithm 1** Road estimation.**Require:** Scene mask of input image, $M_{mask}$**Ensure:** Road mask of input image, $M_{road}$ 1:$M_{trail}$: extract pixel belonging to the trail from $M_{mask}$ 2:$M_{edge}$: extract edge of the trail using canny edge detection algorithm from $M_{trail}$. 3:$Q_{edge}$: get coordinate of edge point from image $M_{edge}$ 4:Let N be number of edge points $Q_{edge}$; 5:Initialize emtpy stack $Q_{road-region}$, and set $Q_{road-region}\left[0\right]$ be the lowest y-coordinate of the $Q_{edge}$; 6:Sort the rest of the edge point of the trail $Q_{edge}$ by polar angle with $Q_{road-region}\left[0\right]$; 7:**for**i=2 to *N*
**do** 8:       $\mathrm{while}\;SIZE(Q_{road-region})>1\;\mathrm{and}\left(Q_{edge}\left[M\right].x-Q_{edge}\left[M-1\right].x\right)*(Q_{edge}\left[i\right].y-$$Q_{edge}\left[M-1\right].y)-\left(Q_{edge}\left[M\right].y-Q_{edge}\left[M-1\right].y\right)*\left(Q_{edge}\left[i\right].x-Q_{edge}\left[M-1\right].x\right)<=0$do 9:          pop $Q_{road-region}$10:        push $Q_{edge}\left[i\right]$ to $Q_{road-region}$11:    **end while**12:**end for**13:$M_{road}$: get the road image based on the cooridinate of the $Q_{road-region}$.14:**return**$M_{road}$

First, the scene mask of trail $M_{trail}$ is extracted based on the color representation of trail, then get the edge of the $M_{edge}$ by applying the canny edge detection algorithm and recording the coordinate of trail $Q_{edge}$. In order to calculate the periphery of trail $Q_{road-region}$, sort the $Q_{road-region}$ by polar angle. Once the $Q_{road-region}$ is sorted, traverse the $Q_{road-region}$. If any coordinate matches the counterclockwise relation with the initial points, consider this edge point $Q_{road-region}$ as the periphery of the trail $Q_{road-region}$. Once the periphery of the trail is estimated, the region is enclosed by the periphery of the trail and can be considered a road region in agroforestry. Note that the grass surrounded by a trail can also be detected as a road region through this method. Moreover, the edge points of the trail are extracted by the scene mask, which means that the accuracy of the scene mask determines the edge points of the trail and influences final result of the road estimation. If the semantic segmentation model generates a bad quality scene mask, the performance of road estimation is also decreased.

## 4. Experiments and Results

### 4.1. Dataset and Experiment Setting

To validate the effectiveness of the proposed ARDformer, we experimented on a large-scale agroforestry dataset, the FreiburgForest dataset [9]. It contains multi-modal images in forested environments. The images were collected at 20 Hz with a 1024 × 768 pixels resolution under three different lighting conditions. The dataset contains over 15,000 images, which correspond to traversing about 14.1 KM. The benchmark includes 366 images with pixel-level ground truth annotation with six classes: Obstacle, Trail, Sky, Grass, Tree and Vegetation. Further annotation was conducted for the road region in this dataset and split the training set with 256 images, validation set with 73 images, and testing set with 37 images. We invited three professional garden workers who have a deep understanding of the drivable areas of agricultural and forestry environments to participate in the labeling. All the experiments are implemented with PyTorch [29] on a single NVIDIA RTX 3090 GPU. We fine-tune our model with training epochs to 10. The initial weight parameter is the pre-trained Image-net parameter, optimizer is AdamW with a 0.0003 learning rate, and soft cross-entropy [30] is used as a loss function.

### 4.2. Performance Metrics

In this paper, we use the intersection of union (IoU) for evaluating our method on metrics. IoU is the ratio of the area of overlap to the area of the union, which indicates the similarity between the predicted region and ground truth. IoU is calculated as follows:(7)$$IoU=\frac{TP_{i}}{TP_{i}+FP_{i}+FN_{i}}$$

Moreover, for evaluating semantic segmentation, overall accuracy (OA), mean intersection over union (mIoU), and F1-score (F1) are chosen as evaluation indices, whose definitions are as follows:(8)$$OA=\frac{\sum_{i=1}^{N}TP_{i}}{\sum_{i=1}^{N}TP_{i}+FP_{i}+TN_{i}+FN_{i}}$$
(9)$$mIoU=\frac{1}{N}\sum_{i=1}^{N}\frac{TP_{i}}{TP_{i}+FP_{i}+FN_{i}}$$
(10)$$Precision=\frac{1}{N}\sum_{i=1}^{N}\frac{TP_{i}}{TP_{i}+FP_{i}}$$
(11)$$Recall=\frac{1}{N}\sum_{i=1}^{N}\frac{TP_{i}}{TP_{i}+FN_{i}}$$
(12)$$F1=2\times \frac{precision\times recall}{precision+recall}$$
where $TP_{i}$, $FP_{i}$, $TN_{i}$ and $FN_{i}$ indicate true positive, false positive, true negative, and false negatives for the specific object indexed as class *i*, respectively.

### 4.3. Performance of Road Detection

As shown in Table 1, our ADRformer outperforms the state-of-the-art rule-based and segmentation-based methods significantly. For a more in-depth analysis of how ADRformer works, we visualize the road detection in agroforestry for a different distribution of trails, and visualization is shown in Figure 5. ADRformer could enclose the trail region and grass surrounded by the trail. Thus, the segmentation-based method’s performance is improved. Compared with the semantic segmentation method, our approach is able to overcome the problem of specific descriptive categories and detect the road based on traversability.

### 4.4. Performance of Segmentation

Since ARDformer is based on segmentation results, we also conducted experiments on the FreiburgForest dataset among SOTA methods, and the results are listed in Table 2. The quantitative indices show the effectiveness of the proposed densely connect hierarchical Transformer. Specifically, our proposed segmentation method achieves 93.34 in mean F1-score, 92.74% in OA, and 84.54% in mIoU, outperforming most of the CNN-based semantic segmentation methods. Benefiting from the global context information model by swin-transformer and hierarchical feature aggregation, the performance of our method is not only superior to the recent SOTA method designed initially for a natural image such as DeepLabV3+ [31] and PSPNet [32], but also prevails over the latest multi-scale feature aggregation models, such as DANet [33] and EANet [34].

### 4.5. Ablation Study on the Hierarchical Transformer

In order to validate the design choice of the proposed densely connected hierarchical transformer, we conducted an ablation study on the FreiburgForest dataset; experimental details are the same as Section 4.1. We select Swin-S [8] as backbone network for ablation study. As shown in Table 3, the hierarchical aggregation module (HAM) fused with the Swin-S yields significant improvement compared to Swin-S baseline in terms of the mIoU. Meanwhile, as feature aggregation can effectively refine feature map by aggregating the different features, utilization of feature aggregation enhances performance dramatically compared with the baseline methods. As shown in Table 3, the increase in mIoU for Swin-S is 3.18%. The integration of Swin-S and HAM achieves the highest accuracy (Table 3).

We also visualized the segmentation result in Figure 6. The red rectangle refers to the significant wrong segmentation region compared to the ground truth. The HAM utilizes the different level feature, and only a fraction of the trails are not segmented, proving that the HAM has a positive contribution to the semantic segmentation. Swin-S correctly segments the scene, but some over-segmentation areas need to be refined by the decoder module. The integration of Swin-S and HAM can deal with the defects of the Swin-S, achieving the best segmentation result.

### 4.6. The Generalization of ADRformer

In order to evaluate the generalization of ARDformer, we performed experiments on another agroforestry dataset, RELLIS-3D [35] and a real agroforestry environment in forests on the outskirts of Beijing, China. As shown in Figure 7b, ARDformer can correctly detect road regions in the RELLIS-3D dataset and real agroforestry environment. In the challenging RELLIS-3D dataset, where a person and shallow water are in the middle of the road, our ADRformer can detect the road region in the scene. For real agroforestry, as shown in Figure 7a, the whole image is much darker than our training set. At the same time, ARDformer can robustly estimate the road region in the real agroforestry environment.

## 5. Conclusions

This study presented a novel two-stage approach for road detection in agroforestry scenes. In order to overcome the limitation of the semantic segmentation method that is not directly interpretable for road detection in agroforestry, we first proposed the densely connected hierarchical transformer for semantic segmentation in agroforestry. Then in the proposed road estimation stage, it enclosed the trail and the grass surrounded by the trail and finally detected the road region. We showed that the two-stage design could further boost the performance of segmentation-based methods, indicating that our two-stage approach is effective in agroforestry. Moreover, extensive experiments on large-scale benchmarks show that our ARDformer outperforms SOTA road detection methods by a large margin. We also showed our ARDformer’s robustness in different datasets and the real world. However, the proposed approach only considers the road’s semantic information and edge curves, not terrain conditions in road detection. If the terrain condition is not considered, the detected road might be risky for UGV. Additionally, the inference speed of our method needs further improvement, and we will accelerate and prune the model to improve the inference speed [36].

In the future, we will consider using LiDAR information [37] to generate the 3D model for road detection and merge it with ARDformer to detect the traversable area in agroforestry. Furthermore, we are considering developing a path planning algorithm design for agroforestry scenes.

## Figures and Tables

**Figure 1 sensors-22-04696-f001:**
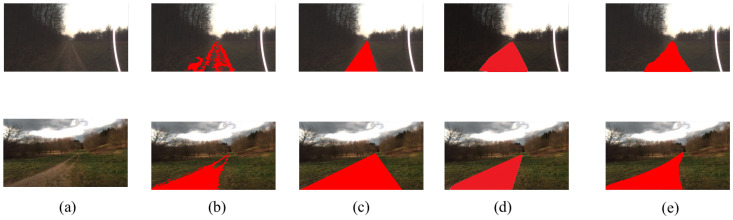
Visualization results of different methods for road detection. Our ARDformer is significantly better than other methods. (**a**) Input image, (**b**) segmentation-based, (**c**) rule-based, (**d**) ARDformer, (**e**) ground truth.

**Figure 2 sensors-22-04696-f002:**
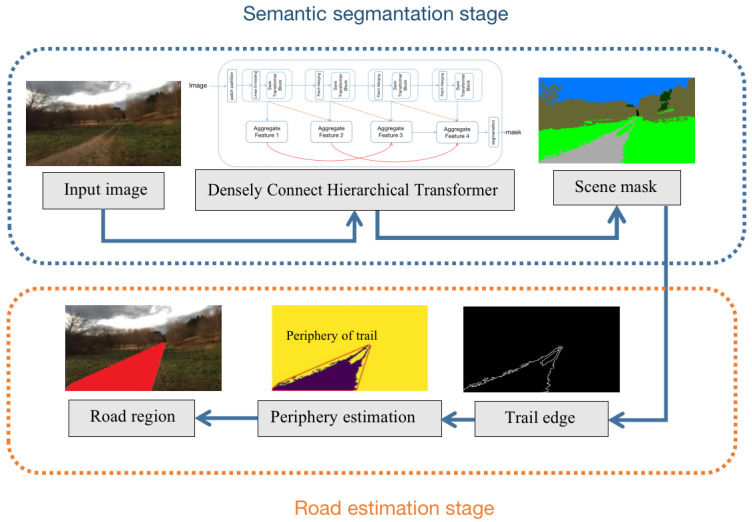
The pipeline of our ARDformer. In the semantic segmentation stage, the scene mask is extracted by a densely connect hierarchical Transformer. In the road estimation stage, the road region is detected based on the periphery information of the trails.

**Figure 3 sensors-22-04696-f003:**
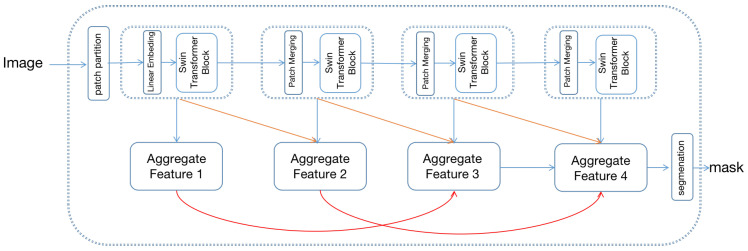
The framework of the proposed densely connect hierarchical Transformer. The top of the image is the swin-transformer and bottom is the hierarchical aggregation module.

**Figure 4 sensors-22-04696-f004:**
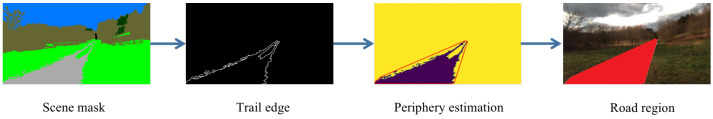
Road detection process using scene mask and edge information. The second image refers to the the edge of the trail and enclosed red line in the third image refers to the periphery of image. The red part in the last image is the road region in agroforestry.

**Figure 5 sensors-22-04696-f005:**
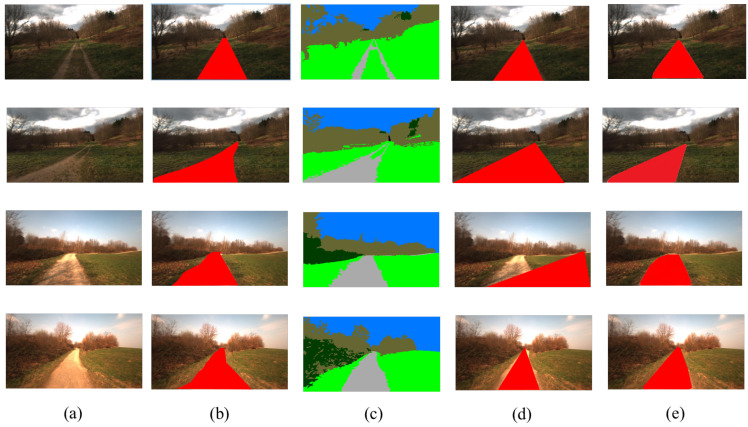
Qualitative results of road detection method. (**a**) Input image, (**b**) ground truth, (**c**) semantic result, (**d**) vanishing point result, (**e**) ARDformer result.

**Figure 6 sensors-22-04696-f006:**
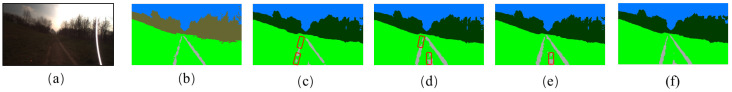
Visualization of ablation study. (**a**) IMAGE, (**b**) Ground Reference, (**c**) ResNet101, (**d**) ResNet101+Feature Aggregation, (**e**) Swin-S, (**f**) SwinS+FeatureAggregation.

**Figure 7 sensors-22-04696-f007:**
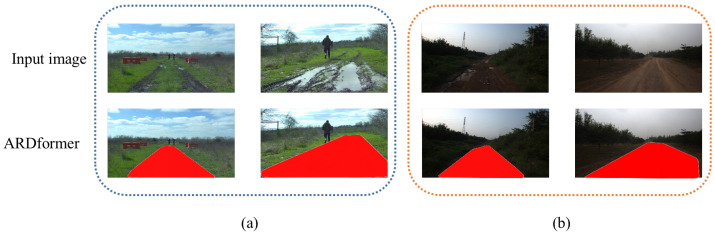
Visualization result of ARDformer in other agroforestry environment. (**a**) RELLIS-3Ddataset, (**b**) real agroforestry environment.

**Table 1 sensors-22-04696-t001:** Comparison with SOTA road detection approach.

Rule Based	Segmentation Based	Ours
75.08	79.47	82.23

**Table 2 sensors-22-04696-t002:** Comparison of model performance on the FreiburgForest dataset. The best performers are highlighted in bold.

Method	*mIoU* (%)	Seg (%)	Seg + Road Estimation (%)
DeepLabV3+	83.15	79.21	80.14
PSPNet	83.78	76.66	77.81
DANet	82.62	78.22	78.37
SegNet	80.67	77.76	78.51
EANet	81.90	81.90	79.83
FCN	77.33	75.29	76.45
Ours	**84.54**	**79.47**	**82.23**

**Table 3 sensors-22-04696-t003:** Ablation study. The swin-transformer [8] with or without HAM. HAM denotes our proposed hierarchical aggregation module (HAM).

Method	Mean *F*1	*OA* (%)	*mloU* (%)
Swin-S	91.72	90.21	81.36
HAM + Swin-S	93.34	92.74	84.54

## Data Availability

The data presented in this study are available on request from the corresponding author.

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
