# Peer review of "ARDformer: Agroforestry Road Detection for Autonomous Driving Using Hierarchical Transformer"

_sensors, 2022, doi:10.3390/s22134696_

Round 1

Reviewer 1 Report

Dear Authors,

   Many thanks for your manuscript submission to MDPI Journal of Sensors. This short paper established a two-stage scheme for road detection in Agro -forestry scenes. The authors claimed that their approach (ARDformer) may outperform other segmentation-based methods, and their experiments on one large-scale benchmark dataset as well as other tests on robustness on different datasets (also real world) suggests the merits of ARDformer. Besides, the authors briefly claimed the limitations of study and future work. I enumerate some potential problems in this research article, I would suggest the authors prepare one round of major revision, and expect the minor revision after passing the next round of double-decision process.

   Major problematic issues suggested to be addressed are listed as below:

   a) Abstract session: While the length of current version is acceptable, I would recommend to extend this paragraph by adding specific details on keynote quantitative scores as concluding remark, which should be more professional. Also, please condense Lines 1-5. The description of your scheme (2nd half of abstract) looks a bit too generic. Please clarify the major steps of your approach and be more specific. 

   b) Introduction: this section cited 15 references and challenged the problems on the technical issues then presented a brief summary on multiple existing methods on road detection. There is a potential defect that the authors almost made nothing on classification of deep learning or prior non-learning based methods, hence, the first few paragraphs are not very clear. I suggest the authors to apply required edits to make this section in a coherent way and smooth the use of English (similar edits can be applied to a second section on Related Work, if consider further extension on the part). In addition, while this section present a short summary on contributions of their work (3 manifolds), the last paragraph is still too generic. Besides, The organization on the remainder of this paper, are expected to be included in the last but separate paragraph. Hence, please consider doing a major rewrite on the last 2-3 paragraphs and be more specific. Thanks a lot!

   c) Methodology (Sections 2): This section basically looks acceptable. The arrangement of images are fine, the resolution and quality of each figure are acceptable. The font size of characters at middle module of top level, must be enlarged. The characters of contexts at Figs. 2-4 should also be zoomed. Besides, I think some statements need adjustment right after the title of this figure in brief descriptions.  

   d) Section 3 (Results and Discussions): Please capitalize the first letter of "Recall" and "Precision" in Equations (11)-(12). Regarding Table 2, I suggest the authors specifically explain why the improvements on per-class IoU or m-IoU are ~5% each on average and less than 10% for best improvement comparing ARDformer to the poorest scheme (i.e., FCN). besides, I think the discussion section should be separate, if not, this section can be renamed as "Experiments and Results". The limitations of study can be explained in the discussion section (along with sensitivity analysis or ablation tests). 

   e) Approach on ARDtransformer and Algorithm: the Algorithm at Page 5 lacks explainations, and the major steps of this part is also unclear. The workflow (architecture) of the proposed transformer lacks explaination of concrete network structures or parameter settings, if the authors missed this problem, please apply the required edits; if not, appreciate for clarification.

   f) Conclusions: this section lacks specific details on limitations of your study, and the last sentence on further study looks generic. The former part needs some supplemental work, and the latter part requires a rewrite (and it is better to be presented in a separate paragraph at the end of this section, including summary of research challenges and proposed future orientations of work plan). Thanks a lot!

   g) References: Citation formats for each references should be updated with respect to the MDPI template on Sensors Journal. Abbreviated terms on the title of journal names and original styles on citing conference proceedings (including the dates and locations) need to be posted. Meanwhile, I think the authors may consider adding more state-of-the-art publications in Years 2018-2022, especially some related MDPI publications to the References (pay attention to latest products in Year 2022). Besides, I think the deep learning based models (not just DNN, any weakly or semi-supervised learning schemes) should be strengthened with respect to the updated introduction part.

   Some minor problematic issues which may require further calibrations:

   a) This paper contains some formatting issues: in the list of authors, those affiliated with "4" are missing. The heading of "References" is also missing. Please make up the missed components and concide with MDPI template.

   b) Apply the same (or similar) font style on the legends and marks of each figure (and subdiagrams), which should be clearly visible. Make sure the size and resolution are fine enough for publication standard. Rearrange the step size of number in y-axis in several subfigures. 

   c) Please stop hyphenating a word that may cross over two adjacent lines (which can be adjusted by applying edits to the MS word template)..

   d) Proofreading is required for the whole context, especially in some long paragraphs in the corresponding Sections.     

   Once again, thank you, and best of luck to your further edits. We will look forward to reviewing your updated manuscript for double-decision.

Best wishes,

Yours sincerely,

Author Response

Dear Reviewer,

We sincerely thank the reviewer for your valuable comments, which greatly help us improve the quality of this manuscript. We revised the manuscript carefully and outlined our responses to the specific comments in the attachment. 

Reviewer 2 Report

The Paper looks good and needs some improvements. 

- The idea is great and should be done for video and not image

- Semantic Segmentation is a hot top but needs to be processed in real time and for video

- Your approach is clearly better and has to be described in detail in a real-time process

- There aren't referenced papers from Sensors Journal

- Math description are weak and need more details

- The algorithm needs to be revised because of some inconsistencies in the description

- I didn't see the final application that your solution fits

- How did you define the performance metrics?

Author Response

(The authors gave the same response as above.)

Round 2

Reviewer 2 Report

Congratulations to the Authors to attend part of the recommendation. You just updated the written. However, I consider it acceptable to be published at this journal.